# *Is Your LLM Secretly a World Model of the Internet*? Model-Based Planning for Web Agents

**Yu Gu**[1][*], **Kai Zhang**[1][*], **Yuting Ning**[1][*], **Boyuan Zheng**[1][*],
**Boyu Gou**[1], **Tianci Xue**[1], **Cheng Chang**[2], **Sanjari Srivastava**[2], **Yanan Xie**[2], **Peng Qi**[2],
**Huan Sun**[1][†], **Yu Su**[1][†]

**[1] The Ohio State University   [2] Orby AI**
**{gu.826, zhang.13253, sun.397, su.809}@osu.edu**

Reviewed on OpenReview: https://openreview.net/forum?id=c6l7yAOHSq

## Abstract

Language agents based on large language models (LLMs) have demonstrated great promise in automating web-based tasks. Recent work has shown that incorporating advanced planning algorithms, *e.g.*, tree search, is advantageous over reactive planning for web agents. However, unlike simulated sandbox environments, real-world environments such as the web are rife with irreversible actions. This undermines the feasibility of backtracking, a cornerstone of (tree) search. Overly relying on test-time search also hurts efficiency. We advocate *model-based planning* for web agents that employs a world model to simulate and deliberate over the outcome of each candidate action before committing to one. We systematically explore this paradigm by: **(1)** Proposing a model-based planning framework, WEBDREAMER, which employs LLMs to serve as both world models and value functions; **(2)** Training specialized LLMs as world models with a scalable data synthesis pipeline. Empirical results demonstrate that WEBDREAMER achieves substantial performance improvements over reactive baselines. It is competitive, while being 4–5 times more efficient, with tree search in sandbox environments (VisualWebArena) and also works effectively on real-world websites (Online-Mind2Web and Mind2Web-Live). Furthermore, our trained world model, Dreamer-7B, performs comparably to GPT-4o, highlighting the potential of specialized world models for efficient and effective planning in complex web environments.[1]

## 1 Introduction

Planning (Mattar & Lengyel, 2022)—deciding on optimal action sequences to achieve goals—has been fundamental to artificial intelligence since its inception. Research into generalist web agents capable of planning and executing a sequence of actions to complete complex tasks across diverse websites has gained significant interest (Deng et al., 2023; Zhou et al., 2024; Zheng et al., 2024; Koh et al., 2024a), partly due to the web's potential as a complex yet realistic environment for driving agent research and development. However, applying existing planning algorithms (Yao et al., 2023a; Hao et al., 2023; Gu et al., 2023; Wang et al., 2025; Feng et al., 2023; Brown et al., 2024, *inter alia*) to the online web environment presents formidable challenges. Real-world environments such as the web are rife with state-changing and irreversible actions—for example, a single website like Amazon.com can involve numerous such actions, including submitting an order, creating an account, changing privacy settings, among many others—making *backtracking*, a cornerstone of search-based planning (Koh et al., 2024b; Putta et al., 2024), highly challenging, if not infeasible. The latency from excessive exploration in test-time search also hurts *efficiency* and compromises user experience.

---

[*]Equal Contribution. See the contribution statement for details.

[†]Equal Advising.

[1]All code, models, and data are publicly available at https://github.com/OSU-NLP-Group/WebDreamer.

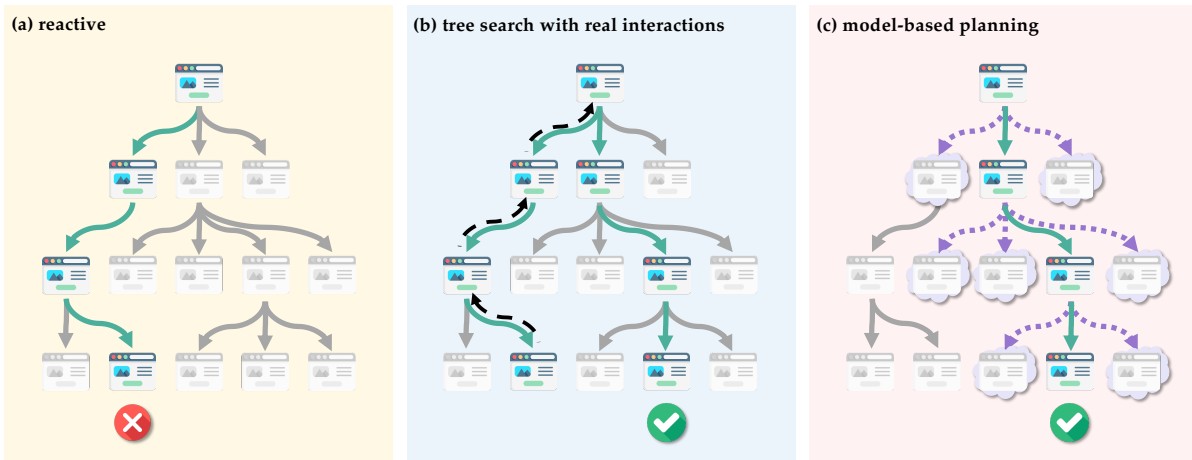

Figure 1: Schematic illustration of different web agent strategies as a search problem, where each node represents a webpage. (a) Reactive: The agent selects locally optimal actions without planning, often leading to suboptimal results. (b) Tree search with real interactions: The agent explores multiple paths via active website navigation, potentially allowing backtracking (dashed arrows). However, backtracking is often infeasible due to irreversible actions. (c) Model-based planning: The agent simulates outcomes (cloud-bordered nodes) before execution, reducing real interactions while maintaining effectiveness.

One promising solution to address these challenges is *model-based planning* (Pascanu et al., 2017; Moerland et al., 2023), which equips agents with the ability to simulate action sequences within a *world model*—a computational representation of environment dynamics. World models have achieved notable success (Ha & Schmidhuber, 2018; Hafner et al., 2020; 2021) in traditional reinforcement learning (RL) tasks within simulated environments like Atari games (Bellemare et al., 2013; Brockman et al., 2016), where environment dynamics are well-defined and the action space is small and fixed, making world model training relatively straightforward. However, building world models for web environments remains under-explored. In contrast to the simulated environments, the Internet is open-ended and ever-evolving, with complex and diverse page structures and a wide range of possible user interactions. This raises the question: *How can we build effective world models for the Internet?*

We propose building world models for the Internet by leveraging large language models (LLMs) as the foundation. Pretrained on web-scale data, LLMs have implicitly acquired both structural knowledge of websites and common sense needed to predict the outcomes of proposed actions, potentially making them well-suited to simulate transitions in complex web environments. To this end, we introduce WEBDREAMER, a model-based planning framework that uses LLMs to simulate and score possible future states before executing actions, thereby enabling informed decision-making, as illustrated in Figure 1. Building on this framework, we further train a dedicated world model, Dreamer-7B, using over 3.1 million interaction instances synthesized by our scalable data synthesis pipeline.

Empirical results demonstrate the effectiveness of our approach: WEBDREAMER, when powered by state-of-the-art LLMs such as GPT-4o, achieves significant performance improvements over reactive baselines across three benchmarks, covering both online and sandbox environments. It is also competitive with the tree search method, while being 4–5 times more efficient on sandbox environment VisualWebArena, and performs effectively on real-world websites (Online-Mind2Web and Mind2Web-Live), where tree search methods are difficult to implement and deploy. Moreover, Dreamer-7B achieves performance comparable to GPT-4o on two online benchmarks. In VisualWebArena, we can continue fine-tuning our Dreamer-7B with in-domain data synthesized by our pipeline. With just 25K training instances, the resulting domain-specific world models yield even stronger results, surpassing GPT-4o. These findings not only demonstrate the potential of using LLMs for model-based planning, but also establish a practical foundation for building world models for the open web through data synthesis, training, and evaluation.

## 2 Related Work

### 2.1 Web Agents

Web agents (Su et al., 2024) powered by (multimodal) language models aim to automate web-based tasks, with benchmarks evolving from MiniWoB++ (Shi et al., 2017; Liu et al., 2018) to WebShop (Yao et al., 2022), Mind2Web (Deng et al., 2023), WebArena (Zhou et al., 2024; Koh et al., 2024a), which introduce more realistic environments and visual challenges. **Reactive Agents** make decisions based on immediate observations without simulation or search (Yao et al., 2023b). Enhancements include prompting proprietary models (Zheng et al., 2024; He et al., 2024; Deng et al., 2023) and training models on HTML and webpage screenshots (Lee et al., 2023; Gur et al., 2023; Furuta et al., 2023; Hong et al., 2024; Baechler et al., 2024). Grounding improvements come from action-coordinate training (You et al., 2024; Cheng et al., 2024; Gou et al., 2025), while human-annotated (Shaw et al., 2023; Hong et al., 2024; Deng et al., 2023; Lai et al., 2024) and synthetic exploration trajectories (Furuta et al., 2023; Song et al., 2024; Patel et al., 2024; Pahuja et al., 2025) further refine agent behavior. However, these agents struggle with multi-step decision-making due to short-sightedness. **Agents with Tree Search** have been explored to enhance decision-making. GPT-4V-based reward modeling (Pan et al., 2024a) and tree search algorithms (Koh et al., 2024b; Putta et al., 2024; Zhang et al., 2024) enable multi-step planning, with variants such as best-first search (Koh et al., 2024b) and Monte Carlo Tree Search (Putta et al., 2024; Zhang et al., 2024). Despite performance gains, search methods significantly increase inference time, face challenges in backtracking on real-world websites, and risk unsafe behaviors like submitting private information to incorrect or unintended web elements.

### 2.2 World Models

World models, central to model-based reinforcement learning (RL; Moerland et al. (2023); Sutton (1991)), learn state transitions to improve sample efficiency (Ha & Schmidhuber, 2018) and support planning (Pascanu et al., 2017; Schrittwieser et al., 2020). Unlike traditional world models in RL focusing on improving data efficiency in the agent learning process, LLM-based world models emphasize decision-making over simulation fidelity, leveraging broad world knowledge for planning (Hao et al., 2023; Kim et al., 2024). Our work extends this line by exploring LLM-based world models in complex web environments. A concurrent work (Chae et al., 2025) also explores augmenting web agents with LLM-simulated action outcomes. However, their focus is on using small-scale data to train *in-domain* world models within sandbox environments, while ours centers on using LLMs and training *general* world models for real-world websites via a scalable data synthesis pipeline. Their sandbox in-domain settings are also discussed in Section 3. Moreover, we evaluate the world model (see Appendix D) and use screenshots as the observation space.

## 3 WebDreamer: Model-Based Planning for Web Agents

### 3.1 Preliminary

Web agents tasked with automating activities in live websites face vast and complex search spaces. Formally, each task, given an instruction $I$, can be formulated as a partially observable Markov decision process (POMDP): $(\mathcal{S}, \mathcal{A}, \mathcal{O}, T, R, \Omega)$, where $\mathcal{S}$ is the set of possible environment states, $\mathcal{O}$ is the set of observations available to the agent, and $\mathcal{A}$ represents actions such as clicking elements, entering text, or navigating URLs. $T : \mathcal{S} \times \mathcal{A} \to \mathcal{S}$ is the state transition function, while $R$ is a binary reward indicating task completion. The agent perceives only an observation $o \in \mathcal{O}$, sampled from the observation function $\Omega(s, a)$.

Tree search-based planning with real interactions is costly and risks irreversible actions. Model-based planning mitigates this by using a learned simulation function $\mathtt{sim} : \mathcal{O} \times \mathcal{A} \to (\mathcal{O} \times \mathcal{A})^*$ which generates imagined trajectories of observations and actions before execution. This enables online planning, where the agent iteratively selects actions based on simulated future trajectories. A common approach is Model Predictive Control (MPC; Garcia et al. (1989)), which simulates future states for each action over a finite horizon $H$, evaluates them using a scoring function $\mathtt{score}$, and executes the action with the highest score. This process repeats after observing new states, allowing adaptive decision-making while avoiding unnecessary interactions.

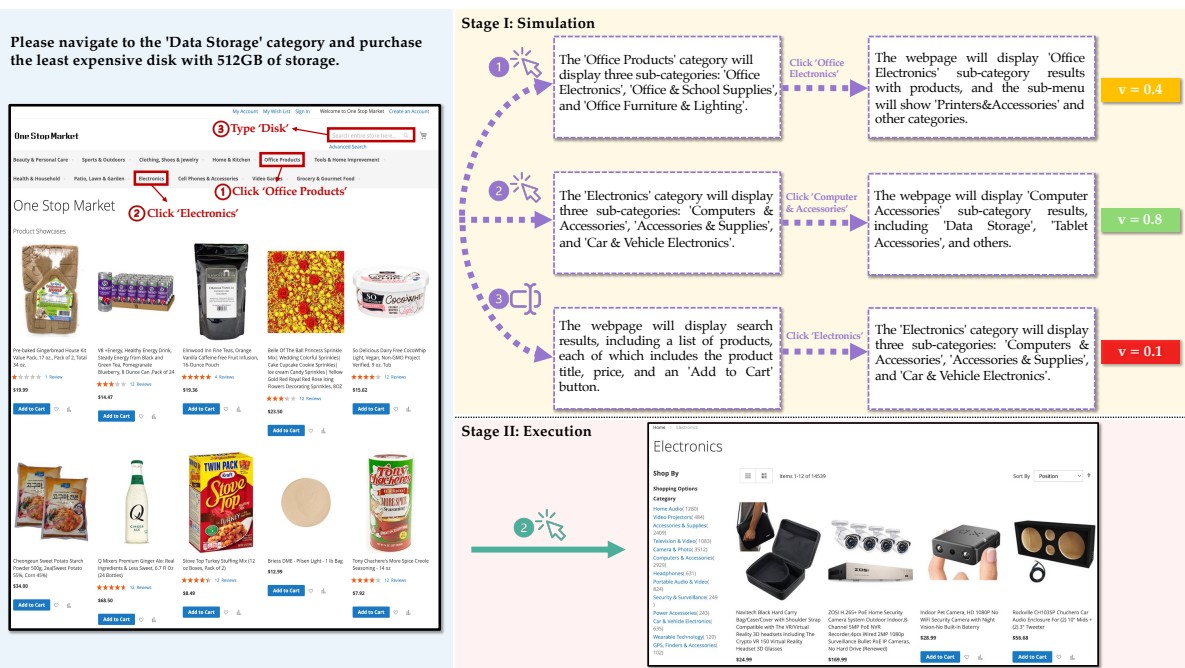

Figure 2: Illustration of WEBDREAMER simulating outcomes for three candidate actions using GPT-4o: (1) *Click "Office Products"*, (2) *Click "Electronics"*, and (3) *Type "Disk" into textbox*. Each dotted box shows an LLM-generated state after a proposed action. Simulated trajectories are scored to identify the best action, and the optimal action *Click "Electronics"* with the highest score ($v = 0.8$) is executed. This example shows a two-step planning horizon. In practice, WEBDREAMER simulates multiple trajectories per action to capture a wider range of possible outcomes, improving coverage and leading to better-informed decisions. Here we only show one trajectory for each action and the final score for brevity.

## 3.2 Core Design

WEBDREAMER follows the planning through simulation paradigm introduced in Section 3.1. Figure 2 illustrates this process with three candidate actions, where WEBDREAMER simulates two-step trajectories for each action, selects the trajectory with the highest score, and executes its corresponding initial action. At its core, WEBDREAMER leverages LLMs as the simulation function (`sim`) and the scoring function (`score`).

**Implementation for `sim`.** Our implementation of `sim` has two modules. (1) A *state-change predictor* approximates the environment transition function $T$ by predicting how the webpage changes after executing an action; this module is *instruction-agnostic*. (2) An *action proposer* then imagines a plausible next action conditioned on the instruction $I$ and the predicted state, enabling long-horizon planning. By alternating between these two steps, `sim` generates trajectories up to depth $H$, where $H$ is a configurable simulation horizon. Concretely, to represent state changes we prompt an LLM (GPT-4o or our self-trained world model) to produce a concise natural language description focused only on the effects of the action (Figure 2, Stage I). The action proposer then uses $I$ and this predicted state to propose the next action (Stage II). In Algorithm 1, we denote this instruction-conditioned rollout succinctly as `sim`$(I, o_t, a)$.

---

**Algorithm 1:** WEBDREAMER

**Input:** Instruction $I$; initial observation $o_0$
**Output:** Sequence of actions $a_0, a_1, \ldots, a_T$
$t \leftarrow 0$;
**while** *True* **do**
    $\mathcal{A}_t \leftarrow$ `get_candidate`$(I, o_t)$;
    $\mathcal{A}'_t \leftarrow$ `self_refine`$(I, \mathcal{A}_t)$;
    $a_t = \arg\max_{a \in \mathcal{A}'_t}$ `score`$(I, $`sim`$(I, o_t, a))$;
    $o_{t+1} \leftarrow$ `execute`$(a_t)$;
    $t \leftarrow t + 1$;
    **if** *termination_check() = True* **then**
        | break;
    **end**
**end**

---

**Implementation for `score`.** After collecting a trajectory $\tau_i$ simulated from each candidate action $a_i$ using `sim`, we further use an LLM as a scoring function for each simulation. Following Koh et al. (2024b), we

prompt GPT-4o to score each simulated trajectory with a three-scale response—complete (1.0), on track (0.5), or incorrect (0)—indicating its progress toward task completion. The final score for each action is averaged over multiple simulated trajectories and scorings, then used to determine the optimal action to execute (*e.g.*, Click "Electronics"), as shown in Stage I of Figure 2.

In addition to `sim` and `score`, a prerequisite to planning is candidate action generation. We employ a two-stage approach: first sampling top-$k$ actions following Koh et al. (2024b), then using an LLM-based `self_refine` module to filter out redundant or implausible candidates before simulation. The role of `self_refine` is to remove unnecessary actions that would lead to near-duplicate or uninformative outcomes. For example, when instructed to *"search PlayStation 5 controller with Deadpool skin"*, the top candidates may include variants such as `type("ps5 controller deadpool")` and `type("play station 5 controller deadpool skin")`, which differ in surface form but lead to nearly identical page states. Simulating all such variants wastes compute without providing additional signal for planning. `self_refine` prompts an LLM to reason over the action set and retain only a concise, diverse subset of candidates that are most likely to affect the trajectory meaningfully. We show the pseudo code of WEBDREAMER's overall design in Algorithm 1. `termination_check` verifies if the model outputs a `stop` action, reaches max steps, or repeats an action over 3 times, also following the implementation by Koh et al. (2024b). Appendix B shows more details.

### 3.3 World Model Data Synthesis and Model Training

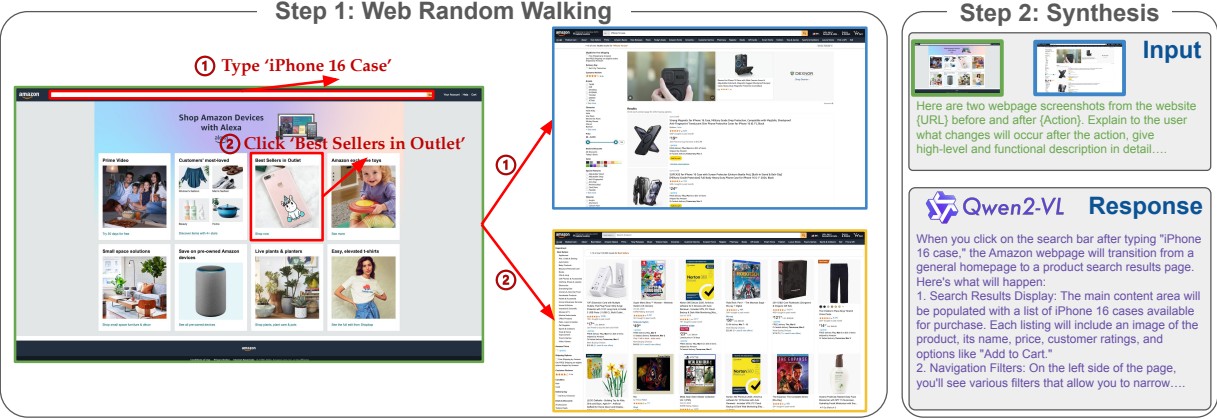

Figure 3: Data synthesis pipeline. The pipeline consists of (1) Web Random Walking, where we autonomously interact with web pages through actions like typing for search and clicking, and (2) Synthesis, where Qwen2-VL-72B generates textual descriptions of state changes based on visual snapshots before and after actions.

While general-purpose LLMs such as GPT-4o have the potential to serve as world models, their cost and latency may limit their feasibility for real-time planning. To explore a more deployable alternative, we also aim to train a small world model that offers lower inference cost and easier adaptation to new domains.

As shown in Figure 3, we develop a scalable data synthesis pipeline that autonomously interacts with web pages using lightweight heuristic guidance. The pipeline requires only a pool of seed URLs (sampled from the October 2024 Common Crawl Index[2]) and can be run in parallel to scale to millions of interactions without human supervision. At each step, the raw HTML is parsed to enumerate feasible actions, such as clicking elements, hovering, typing into text boxes, or selecting options. To faithfully align these actions with the visual modality (screenshot), we maintain a mapping between each HTML element and its corresponding location on the rendered screenshot, ensuring that actions identified from the HTML can be accurately represented in the visual action descriptions.

To better reflect real usage patterns, we sample actions with fixed probabilities that favor frequent interactions like clicking, while still ensuring sufficient coverage of less common operations (*e.g.*, selecting options). Furthermore, to encourage causal dependencies across steps, we explicitly track HTML changes after each

---

[2]https://commoncrawl.org

action and increase the probability of acting on newly revealed elements. For instance, after hovering over a menu tab, the pipeline prioritizes clicking items in the newly expanded dropdown (with 70% probability), while still allowing occasional actions on previously visible elements. For typing actions, we generate contextually relevant queries using GPT-3.5-Turbo to ensure realistic search interactions.

Once an interaction is performed, we capture visual snapshots before and after the action. We then prompt Qwen2-VL-72B (Wang et al., 2024) to generate textual descriptions detailing the changes in the webpage state (Figure 3 Step 2), ensuring an accurate representation of how each action impacts the visual content. Each training instance consists of the initial visual state, the action taken, and the generated textual description of the state change. After data collection, we filter out failed interactions, automation-blocked content, and potentially harmful data, resulting in a final dataset of over 3.1M interaction instances that capture rich causal relationships between user actions and web state transitions.

As we empirically find horizon step $H=1$ to be the most effective and efficient configuration, we focus on training the state transition function in sim, initializing it with Qwen2-VL-7B (Wang et al., 2024). The final model, Dreamer-7B, is trained to predict the next state as a natural language description after performing an action on the current state, using a next-token prediction objective. To efficiently monitor the progress of self-trained world models without relying on costly downstream evaluations for every checkpoint, we construct an intrinsic evaluation set for checkpoint selection, detailed in Appendix D. Appendix C provides additional details on data synthesis and training.

## 4 Experiments

### 4.1 Setup

To properly test our planning framework's real-world performance, we focus on three representative web agent benchmarks, capturing the dynamic nature of web interactions: **VisualWebArena** (VWA; Koh et al. (2024a)) is designed to evaluate multimodal agents in visually grounded tasks. It includes 233 tasks verified by humans across three self-hosted websites: Classifieds, Shopping, and Reddit. The metric success rate is calculated as the percentage of tasks successfully accomplished by the agent-generated trajectories. **Online-Mind2Web** (Xue et al., 2025) is an online benchmark derived from Mind2Web (Deng et al., 2023), including 300 updated or newly created high-quality tasks spanning 136 real-world websites. These tasks can be categorized into easy, medium, and hard based on the number of steps required for completion. To reduce cost, we use a subset of 100 tasks, randomly sampling 30 easy, 40 medium, and 30 hard tasks. The benchmark employs an automatic evaluation pipeline to measure task success rate, achieving an 85% agreement with human judgment. **Mind2Web-Live** (Pan et al., 2024b) consists of 104 tasks in 69 real-world websites refined from Mind2Web (Deng et al., 2023). It defines and annotates critical intermediate steps as key nodes for each task and considers a task successful only if all key nodes are completed.[3] For all benchmarks, we use screenshots as the observation space and add Set-of-Mark (Yang et al., 2023) in VWA for fair comparison with the tree search baseline. In our experiments, we empirically set the planning horizon $H$ to 1. A comprehensive analysis of this parameter is presented in Section 5.1.

To demonstrate the effectiveness of our framework and trained world models, we primarily compare with two baselines: a reactive agent and a tree search agent with real interactions.[4] The reactive baseline uses GPT-4o as the policy model. Given the task instruction, current observation, and action history, it predicts the next action directly, without any simulation, search, or self-refinement. While the LLM may internally perform chain-of-thought reasoning (Wei et al., 2022), the agent is reactive because it chooses only the immediate next step without explicit lookahead planning. We adopt the official GPT-4o with Set-of-Mark (Yang et al., 2023) implementation from the VWA codebase and adapt it for Online-Mind2Web and Mind2Web-Live. For the tree search baseline (Koh et al., 2024b), we report results only on VWA, as performing real-interaction tree search is infeasible on real-world websites in Online-Mind2Web or Mind2Web-Live. In VWA, Koh et al. (2024b) restore previous states by resetting the sandbox environment and re-executing the corresponding

---

[3]We ensure no overlap between Online-Mind2Web and Mind2Web-Live for better task diversity.
[4]For brevity, we refer to tree search with real interactions simply as tree search in our experiments.

action sequences, a mechanism unavailable on real websites. This makes tree search unsuitable beyond VWA, whereas our WEBDREAMER readily applies to all benchmarks.

## 4.2 Main Results

| Method | World Model | VisualWebArena | Online-Mind2Web | Mind2Web-Live |
|---|---|---|---|---|
| Reactive | - | 17.6 | 26.0 | 20.2 |
| Tree Search | - | **26.2** | - | - |
| WEBDREAMER | GPT-4o | 23.6 | **37.0** | **25.0** |
| | Qwen2-VL-7B | 17.2 | 31.0 | 19.2 |
| | Qwen2-VL-72B | 21.0 | 31.0 | 18.3 |
| | Dreamer-7B | 21.9 | 35.0 | 24.0 |

Table 1: Success rate (%) on VisualWebArena (Koh et al., 2024a), Online-Mind2Web (Xue et al., 2025), and Mind2Web-Live (Pan et al., 2024b). We implement all the baselines ourselves to avoid discrepancies due to hardware and experimental settings in prior works.

**Effectiveness.** We present the overall performance results in Table 1. WEBDREAMER demonstrates substantial improvements over the reactive agent on all benchmarks. Notably, on the VWA dataset, our proposed method achieves a 34.1% relative performance gain and only trails behind tree search slightly. It is important to note that tree search is not very practical on real-world websites, whereas WEBDREAMER provides a more flexible and adaptive alternative. On Online-Mind2Web and Mind2Web-Live, WEBDREAMER outperforms the reactive baseline by a relative gain of 42.3% and 23.8%, respectively. The strong results show the effectiveness of WEBDREAMER across different real-world websites.

Secondly, training world models on our large-scale synthesized data proves to be effective. As shown in Table 1, fine-tuning Qwen2-VL-7B into Dreamer-7B leads to a substantial 4.7% absolute improvement in success rate on VisualWebArena, 4.0% on Online-Mind2Web, and 4.8% on Mind2Web-Live, outperforming the vanilla Qwen2-VL-7B model and even Qwen2-VL-72B. Furthermore, Dreamer-7B achieves performance comparable to GPT-4o on two online benchmarks, Online-Mind2Web and Mind2Web-Live, demonstrating the feasibility of training world models for web-based decision-making.

**Efficiency.** Another key advantage of model-based planning is its efficiency compared with tree search using actual explorations. Table 2 shows that tree search requires approximately 3 times more steps than the reactive baseline, whereas our method maintains comparable number of action steps. Notably, compared to reactive baselines, tree search introduces about 10 times greater latency (in wall clock time) due to additional actions and backtracking, while the overhead from simulation in our approach is substantially lower, making WEBDREAMER 4–5 times more efficient than the tree search baseline.

| Steps | Reactive | Tree Search | WebDreamer |
|---|---|---|---|
| Classifieds | 3.4 | 9.9 | 4.1 |
| Reddit | 5.1 | 13.6 | 5.2 |
| Shopping | 4.5 | 11.4 | 4.5 |

| Seconds | Reactive | Tree Search | WebDreamer |
|---|---|---|---|
| Classifieds | 68.3 | 749.2 | 183.6 |
| Reddit | 83.5 | 972.1 | 233.7 |
| Shopping | 87.7 | 785.7 | 179.4 |

(a) Number of action steps.                    (b) Task completion wall clock time.

Table 2: Efficiency analysis on VWA. All methods here use GPT-4o for fair comparison.

# 5 Discussions

## 5.1 Planning Framework

**Ablation.** We perform ablation studies on the simulation and self-refinement stages of WEBDREAMER on the VWA shopping human subset, which is the largest subset verified by humans. We pay special attention to the simulation stage, which is the core of model-based planning. One might argue that the primary improvement stems from reranking candidate actions, irrespective of whether this ranking relies on simulation. To test this idea, we conduct an experiment where we remove the simulation stage

and instead ask the reward model (`score`) to directly evaluate each candidate action (Reranking). Additionally, we remove the self-refinement step after the action proposal in our framework to assess its impact (*w/o* Self-Refinement).

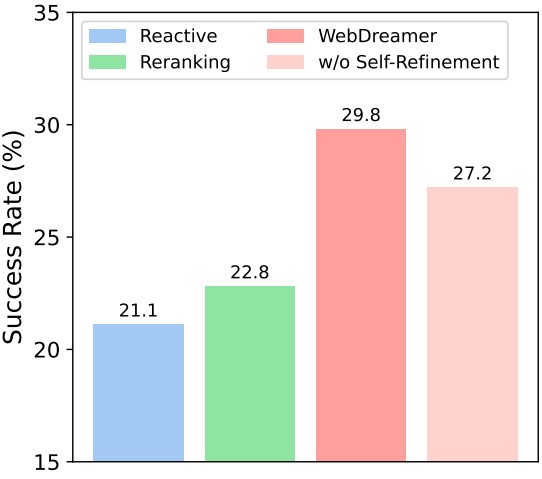

Figure 4: Ablation study on the simulation stage and self-refinement stage.

Upon closer examination, we find that this decline is primarily due to the self-refinement module's ability to effectively filter out less relevant candidate actions when the next optimal action is clear. In contrast, directly simulating all actions may introduce additional noise that can negatively impact performance. As shown in Figure 4, this modified reranking approach does lead to some improvement over the reactive baseline, but the gain is small and still falls well behind WEBDREAMER. These results confirm that the LLM-based world model simulation plays a crucial role in the planning process. Furthermore, we observe a decrease in performance when removing the self-refinement stage.

**Planning Horizon.** As introduced in Section 3.2, WEBDREAMER supports configurable planning horizon $H$ (*i.e.*, the simulation depth). To gain deeper insights into its effectiveness and current limitations, we investigate how the planning horizon affects the final performance. Using GPT-4o as the world model, we evaluate WEBDREAMER with planning horizons of 1, 2, and 3 on the same subset of Online-Mind2Web.

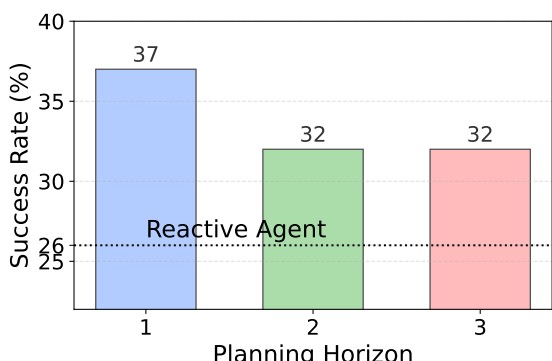

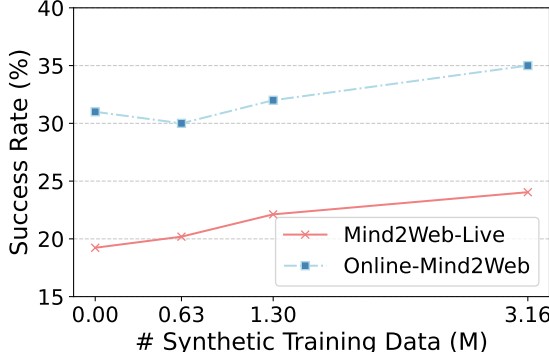

Figure 5: Performance of WEBDREAMER with GPT-4o and different planning horizons $H$ on Online-Mind2Web.

Figure 6: Performance of WEBDREAMER with Dreamer-7B w.r.t. different sizes of training data on two online benchmarks.

As depicted in Figure 5, the performance consistently outperforms the reactive baseline across all horizon settings. Nonetheless, when the planning horizon extends to 2 and 3 steps, the effectiveness begins to diminish. Upon closer examination, this is primarily due to action proposal hallucinations within the simulation. Specifically, the action proposal within simulation is biased toward generating seemingly relevant actions for task completion, even when these actions are not available based on the predicted outcome. As a result, as the planning horizon increases, the trajectories simulated from different actions become less distinguishable, as

they all appear somewhat correct.[5] Given the complexity of the web environment, simulating multiple steps ahead is challenging due to error accumulation, which aligns with previous observations (Mendes & Ritter, 2025; Chae et al., 2025). However, achieving better performance with longer horizons is not a major goal of this work; instead, we aim to show the feasibility and potential of using LLMs for model-based planning. We leave this as a venue for future improvement.

## 5.2  Training World Models

**Scaling Trend.**  We investigate the scaling trend by gradually increasing the size of our synthetic training data, as shown in Figure 6. For Mind2Web-Live, performance steadily improves with larger training datasets, though the rate of improvement tapers off at higher data scales, suggesting potential diminishing returns. In contrast, Online-Mind2Web shows more consistent gains overall, following a slight performance drop with small-scale training data. These observations suggest that further scaling may continue to yield performance improvements. We omit scaling results on the sandbox environments in VWA and focus our analysis on the other two online benchmarks, which more accurately reflect real-world environments and web agent tasks.

**In-Domain Continual Training.**  For specific environments, we can synthesize domain-specific data for world model training, enabling more specialized and contextually grounded simulations. In VWA (Koh et al., 2024a), we employ our data synthesis pipeline introduced in Section 3.3 to synthesize 25K in-domain interactions for each of the three environments: Classifieds, Reddit, and Shopping. To prevent test data leakage, we filter out search actions containing queries that appear in test examples. The final in-domain checkpoints are continually trained from the Dreamer-7B model, resulting in three separate world models, each specialized for its respective environment.

|  |  | Classifieds | Reddits | Shopping | Total |
|---|---|---|---|---|---|
| Reactive |  | 17.9 | 14.3 | 19.3 | 17.6 |
| Tree Search (Koh et al., 2024a) |  | **26.8** | **20.6** | **28.9** | **26.2** |
| WebDreamer | GPT-4o | 23.2 | 17.5 | 26.3 | 23.2 |
|  | Qwen2-VL-7B | 17.9 | 11.1 | 20.2 | 17.2 |
|  | Qwen2-VL-72B | 19.6 | 15.9 | 24.6 | 21.0 |
|  | Dreamer-7B | 21.4 | 15.9 | 25.4 | 21.9 |
|  | + In-Domain | 25.0 | 15.9 | 26.3 | 23.2 |

Table 3: Success rate (%) of WEBDREAMER with various world models on VWA.

As shown in Table 3, continual training improves performance over the Dreamer-7B model and achieves results comparable to or even better than GPT-4o in certain environments. The Classifieds and Shopping domains benefit the most from in-domain adaptation, demonstrating that domain-specific fine-tuning successfully refines model predictions to better reflect the specific environment dynamics. However, performance on Reddit remains unchanged, likely due to its dense viewport and limited functional interactions. Unlike Classifieds and Shopping, which have simpler visual organization, Reddit pages contain long, text-heavy content, making it difficult for a world model to infer meaningful state changes beyond surface-level text visibility. In addition, the limited perception abilities (Gou et al., 2025; Zhang et al., 2025) of 7B models in dense web viewports may further constrain them to simulate fine-grained changes in text-heavy environments. These results underscore the effectiveness of domain-specific adaptation while highlighting areas for further improvement in specific web domains. Future work can explore representation techniques to better handle dense web layouts, further expanding the applicability of world models across diverse real-world web environments.

**Case Studies.**  We present an example to explore the differences between GPT-4o and Dreamer-7B as world models. GPT-4o provides more detailed state change descriptions and considers multiple possible outcomes, though some of these details are unimportant or irrelevant. For example, as shown in Figure 7, GPT-4o

---

[5]We observed a slight improvement from planning horizon 1 to 2 on VWA (Appendix E.1), but chose horizon 1 for our main experiments as it performs comparably to horizon 2 while being significantly less costly.

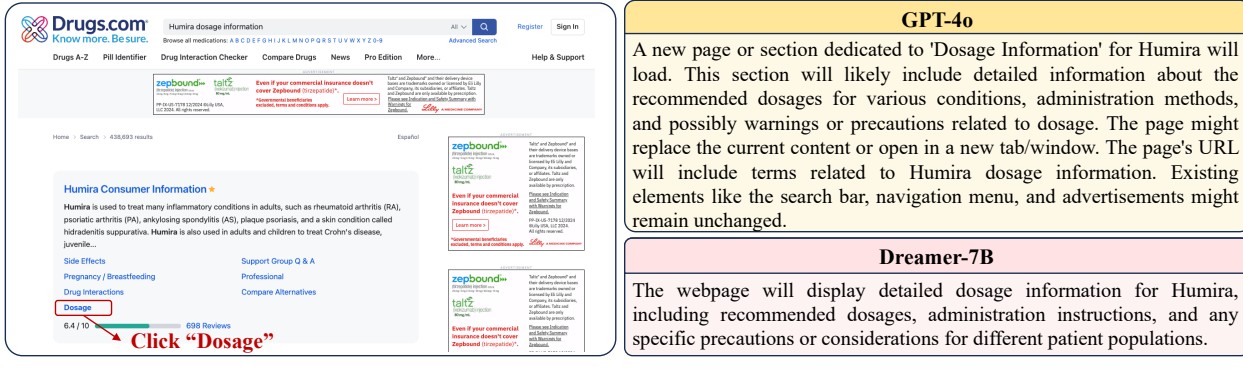

Figure 7: An example comparing GPT-4o and Dreamer-7B as world models. General-purpose models such as GPT-4o often provide detailed but sometimes noisy predictions, while our Dreamer-7B focuses on concise, action-relevant outcomes.

successfully predicts the key state changes after clicking on "Dosage," outlining the relevant information that may appear on the new webpage. However, its prediction also includes unimportant details such as URL and UI changes, which may introduce noise for the policy model. On the contrary, Dreamer-7B offers more concise and action-oriented predictions, offering the most important information on the new webpage.

To further clarify the role of simulation in planning, we also present case studies with GPT-4o as a world model, covering both positive and negative examples in Appendix F. They illustrate how simulation aids the agent in exploring the environment, as well as how inaccuracies in simulation can lead to incorrect predictions.

## 6 Conclusion

In this paper, we systematically explore the use of LLMs as world models for model-based planning in web environments. Our planning framework, WEBDREAMER, achieves substantial improvements over reactive baselines across three benchmarks and offers a 4–5 times more efficient yet competitive alternative to tree search, which is often impractical due to backtracking and efficiency constraints on real-world websites. Beyond leveraging off-the-shelf proprietary LLMs as world models, we train Dreamer-7B on 3.1 million web interaction examples synthesized through our scalable data generation pipeline, achieving performance comparable to GPT-4o. Continual in-domain fine-tuning further allows Dreamer-7B to adapt to specific environments, improving simulation quality and downstream performance, even surpassing GPT-4o. This work lays the foundation for future research on model-based planning for efficient and effective decision-making in web environments and establishes a path toward scaling world models to handle more general web tasks. Overall, our results suggest that the answer to our title question is yes: LLMs can serve as effective world models for web planning, though there remains significant room for improvement in multi-step simulation.

## Ethical Considerations

Our work focuses on model-based planning for web agents by using and training world models on web environments. Although the framework is developed for research purposes, we acknowledge potential risks if misused. For example, automated web agents could be misused for spamming, excessive scraping, or other harmful activities, and releasing interaction data without safeguards could raise concerns about privacy or misuse. To mitigate these risks, we filter collected data to remove failed or harmful interactions, and no personally identifiable information or explicit content is retained. Our contributions are intended solely for academic research, and we caution against deploying our system in ways that compromise user privacy, security, or website integrity. We strongly encourage responsible use of this research within appropriate ethical and legal boundaries.

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

## Overview

The appendix includes the following sections:

## A Contribution Statement

**Yu Gu** conceived the project with Yu Su and developed the planning framework WEBDREAMER. He implemented the core codebase, led the design and execution of key experiments including ablation studies, and wrote the initial manuscript draft. He also led the public release of the resources.

**Kai Zhang** led the data synthesis and world model training efforts, including the training of Dreamer-7B and domain-specific variants. He conducted experiments on VisualWebArena and played a major role in editing and polishing the manuscript to its final form.

**Yuting Ning** led the development of the intrinsic evaluation suite and conducted experiments on Online-Mind2Web and Mind2Web-Live. She also contributed significantly to the final manuscript refinement.

**Boyuan Zheng** contributed to the development of the planning framework, the data synthesis pipeline, and the in-domain world model training. He also provided valuable insights into the experimental design.

**Orby AI team** (Cheng Chang, Sanjari Srivastava, Yanan Xie, and Peng Qi) provided large-scale compute resources and data crawling support, which were essential for scaling up the data synthesis and training pipeline. Peng Qi and Yanan Xie also provided constructive feedback throughput the project.

**Yu Su** and **Huan Sun** steered the main directions throughout the project, led discussions on the planning framework and experiments, and provided funding support. **Yu Su** conceived the project with Yu Gu.

All authors reviewed the manuscript and provided feedback.

## B Prompts for Four Stages in WebDreamer

### B.1 Action Proposal

> **Action Proposal**
>
> You are an autonomous intelligent agent tasked with navigating a web browser. You will be given web-based tasks. These tasks will be accomplished through the use of specific actions you can issue.
> Here's the information you'll have: {`Web Information`}
> The user's objective: {`Task Objective`} This is the task you're trying to complete.
> The current web page screenshot: {`Web Page Screenshot Image`} This is a screenshot of the webpage, with each interactable element assigned a unique numerical id. Each bounding box and its respective id shares the same color.
> The observation, which lists the IDs of all interactable elements on the current web page with their text content if any, in the format `[id][tagType][text content]`. `tagType` is the type of the element, such as button, link, or textbox. `text content` is the text content of the element. For example, `[1234][button]['Add to Cart']` means that there is a button with id 1234 and text content `'Add to Cart'` on the current web page. `[][StaticText][text]` means that the element is of some text that is not interactable.

The current web page's URL: {`Web URL`} This is the page you're currently navigating.
The open tabs: {`Previous Tabs`} These are the tabs you have open.
The previous action: {`Previous Action`} This is the action you just performed. It may be helpful to track your progress.
The actions you can perform fall into several categories:

Page Operation Actions:
  - `click [id]`: This action clicks on an element with a specific id on the webpage.
  - `type [id] [content]`: Use this to type the content into the field with id. By default, the `Enter` key is pressed after typing unless `press_enter_after` is set to 0, *i.e.*, `type [id] [content] [0]`.
  - `hover [id]`: Hover over an element with id.
  - `press [key_comb]`: Simulates the pressing of a key combination on the keyboard (*e.g.*, Ctrl+V)
  - `scroll [down]` or `scroll [up]`: Scroll the page up or down.

Tab Management Actions:
  - `new_tab`: Open a new, empty browser tab.
  - `tab_focus [tab_index]`: Switch the browser's focus to a specific tab using its index.
  - `close_tab`: Close the currently active tab.

URL Navigation Actions:
  - `goto [url]`: Navigate to a specific URL.
  - `go_back`: Navigate to the previously viewed page.
  - `go_forward`: Navigate to the next page (if a previous `go_back` action was performed).

Completion Action:
  - `stop [answer]`: Issue this action when you believe the task is complete. If the objective is to find a text-based answer, provide the answer in the bracket.

Homepage:
If you want to visit other websites, check out the homepage at http://homepage.com. It has a list of websites you can visit. http://homepage.com/password.html lists all the account name and password for the websites. You can use them to log in to the websites.

To be successful, it is very important to follow the following rules:
  1. You should only issue an action that is valid given the current observation
  2. You should only issue one action at a time.
  3. You should follow the examples to reason step by step and then issue the next action.
  4. Generate the action in the correct format. Start with a *"In summary, the next action I will perform is"* phrase, followed by action. For example, *In summary, the next action I will perform is* `click [1234]`.
  5. Issue stop action when you think you have achieved the objective. Don't generate anything after stop.

## B.2 Self-Refinement

### Self-Refinement

You are assisting a web navigation agent to help a human user navigate a website to complete a task. Given the user's intent, the action history, and the current state of the webpage, the agent has proposed a set of candidate actions to take at the current step.

Your role is not to determine a best action for the agent at this step, but to filter out the actions that are very likely not relevant or helpful for the agent to accomplish the task.

Please select all actions that you think that could possibly lead the agent to accomplish the task. It's important to note that to accomplish a task, the agent will execute a sequence of actions. So the action to take at this step does not have to immediately lead to the completion of the task. You should select any action that could be relevant for the agent to take in the current state of the webpage. Try to be as thoughtful and comprehensive as you can! Don't miss any possible action. If there is one action that is clearly the best, and all other actions are clearly not very relevant, you can only select one action. Please do this sparely, since some actions may be helpful in a longer horizon.

An action should be included as long as it could be relevant to the task, even if it may not be the most direct action to take at this step!! Some relevant actions might seem indirect at the first glance, but could be helpful in a longer horizon. Please also include those actions.

Please at least select one action.

**\*IMPORTANT\***
Format your response into two lines as shown below:

Thoughts: `<your thoughts and reasoning process>`. You must explicitly evaluate each action one by one and imagine whether it could be relevant to the task following the format: `action:...  rationale:...`

Selected actions: `id0;id1;aid2;...` (please return the index of the action in the candidate actions list, starting from 0. Don't output the action description itself. Separate the indices with semicolons. Do not add spaces or any other characters after the semicolons.)

Action History: {`last_actions_str`}

Current URL: {`current_url`}

The images corresponding to the user intent are shown in the FIRST {`len(intent_images)`} images (before the User Intent).

The last {`len(screenshots)`} snapshots of the agent's trajectory are shown in the LAST {`len(screenshots)`} images. The LAST IMAGE represents the current state of the webpage.

Proposed Action: {`action_descriptions`}

## B.3  World Model

### World Model (state transition function)

You are an agent that predicts the effect of an action on a webpage. You will be given a screenshot of a webpage, a sequence of actions and state changes applied to the initial screenshot, and an operation to perform on the webpage. You are required to predict the new changes that will occur on the webpage after the operation is performed, such as the appearance of new elements, the disappearance of existing elements, or changes in the content of existing elements. The operation type and the element to operate will be provided in the prompt. Directly output `State changes:...` and don't output anything else. Try to be as comprehensive and detailed as possible.

Based on the initial screenshot and the changes to the webpage, please predict the changes after action:

## B.4  Reward Model

### Reward Model (`score`)

You are an expert in evaluating the performance of a web navigation agent. The agent is designed to help a human user navigate a website to complete a task. Given the user's intent, the agent's action history, the current state of the webpage, your goal is to decide **whether the simulated steps by the agent indicate a successful execution of the user intent**. In particular, if the predicted state (*i.e.*, the current state represented by the last image plus all the predicted changes so far) corresponds to a successful final state. If it is a failure but it looks like the simulated steps are on the right track towards success, you should also output as such. Note that, in the simulated steps, all the state changes are predicted by the agent's world model, and they may not actually be faithful to the real website interactions (*e.g.*, some proposed actions may not be available in a realistic website). You should also account for this in your evaluation (*e.g.*, if the predicted state changes are not reasonable then it's probably a failure).

> **\*IMPORTANT\***
>
> Format your response into two lines as shown below:
>
> Thoughts: `<your thoughts and reasoning process>`
> Status: `"success"` or `"failure"`
> On the right track to success: `"yes"` or `"no"`

## C   Data Synthesis and World Model Training

### C.1   Data Synthesis Prompt

| Prompt Templates |
| --- |
| Here is the web screenshot <image_token>. Please describe what you would see after performing {`action`} on {`element_description`}. |
| Here is the web page you are looking at <image_token>. Please describe what you would see after doing {`action`} on {`element_description`}. |
| Here is the web page you are currently at <image_token>. Describe what you will see after {`action`}ing {`element_description`}. |
| Here is the current web page <image_token>. Briefly describe what you will see after {`action`}ing {`element_description`}. |
| Below is the current screenshot <image_token>. Briefly describe what you will see after {`action`}ing {`element_description`}. |
| Below is the current screenshot <image_token>.  Describe what you will see after {`action`}ing {`element_description`}. |
| Below is the current screenshot <image_token>. Please describe what you would see after a {`action`} on {`element_description`}. |

Table C.1: Prompt templates used to generate language descriptions of state transitions.

In practice, we first draw a red bounding box around the target element to precisely localize it for Qwen2-VL-72B. Next, we prompt Qwen2-VL-72B to separately describe the element using a referring expression and to describe the resulting state change. Finally, we combine the referring expression and the state change description to construct the training instance, using one of the templates randomly selected from Table C.1. We list the prompt used to synthesize natural language descriptions of the next state below.

Despite the fact that our training data includes only a few prompt templates and natural images, experiments in Section 4 have shown that the model generalizes well to unseen instruction or prompts used in benchmarks like Online-Mind2Web (Xue et al., 2025) and Mind2Web-Live (Pan et al., 2024b) and to images with Set-of-Mark (Yang et al., 2023) in VisualWebArena (Koh et al., 2024a).

> **Data Synthesis**
>
> Here are two webpage screenshots from the website {URL} before and after {Action} on the element within the red bounding box.
>
> **Element Description:** Please describe the element within the bounding box to ensure user can locate this element in the webpage image only using this description (only element description, not bounding box). So DO NOT say something like the element within the bounding box, DO NOT SAY anything about red bounding box.
> Instead, describe the element with referral expression like the button showing the text 'Make Appointment' or the 'Search' in the search bar.
> Starts with lower case and make sure the description is a noun like the element that is a category link labeled 'Massage' located in the sidebar on the left side of the page.
>
> **Change Description:** Explain to the user what changes will occur on the webpage after they click on the described element. Do not say too many trivial details. Instead, give high-level and functional description in detail after the action. Focus solely on describing the changes that will happen, not the element.

## C.2   Training Data Statistics

|             | Number    | Percentage |
|-------------|-----------|------------|
| Unique URLs | 1,247,960 | -          |
| Action      |           |            |
| - Click     | 2,653,704 | 84.0       |
| - Hover     | 241,234   | 7.6        |
| - Type      | 217,692   | 6.9        |
| - Select    | 47,617    | 1.5        |
| Total       | 3,160,247 | 100        |

Table C.2: Training data distribution.

As described in Section 3.3, we develop a scalable pipeline to synthesize large-scale interaction data by randomly exploring web pages across a wide range of domains. This process results in over **3.1 million** interaction instances spanning **1.2 million unique URLs**, as summarized in Table C.2. We focus on four primary interaction types: *click*, *hover*, *type*, and *select*, which collectively reflect the core user intents on modern web interfaces. Clicks dominate the dataset (84%), consistent with their central role in triggering state transitions on the web. Although our data synthesis pipeline initially included *scroll* actions, we empirically found that they contributed little to downstream performance and thus excluded them from the final training set.

Importantly, the interactions in our dataset are not restricted to initial steps but span various stages within multi-step trajectories. Figure C.1 shows the distribution of actions by their position in the interaction sequence. The average position is 4.4, and a substantial portion of actions occur deeper in the trajectory. This distribution allows the world model to learn more informative web dynamics and have the potential to enable long-term planning.

## C.3   World Model Training

All world models are fine-tuned using the Qwen2-VL-7B-Instruct (Wang et al., 2024) backbone. To align with future state prediction, we format training examples using structured prompts, such as: *"Here is the web screenshot. Please describe what you would see after performing {action} on {element}."* All experiments are conducted on 64 H100 GPUs with 80GB memory each. The final Dreamer-7B model is trained for up to 2 epochs over the full training dataset described above. We evaluate models every 1000 steps using an intrinsic evaluation metric (described later) to ensure fair and consistent model selection across settings. We use the DecoupledAdamW (Loshchilov & Hutter, 2019) optimizer with a learning rate of `1e-6`, $\beta_1 = 0.9$, and

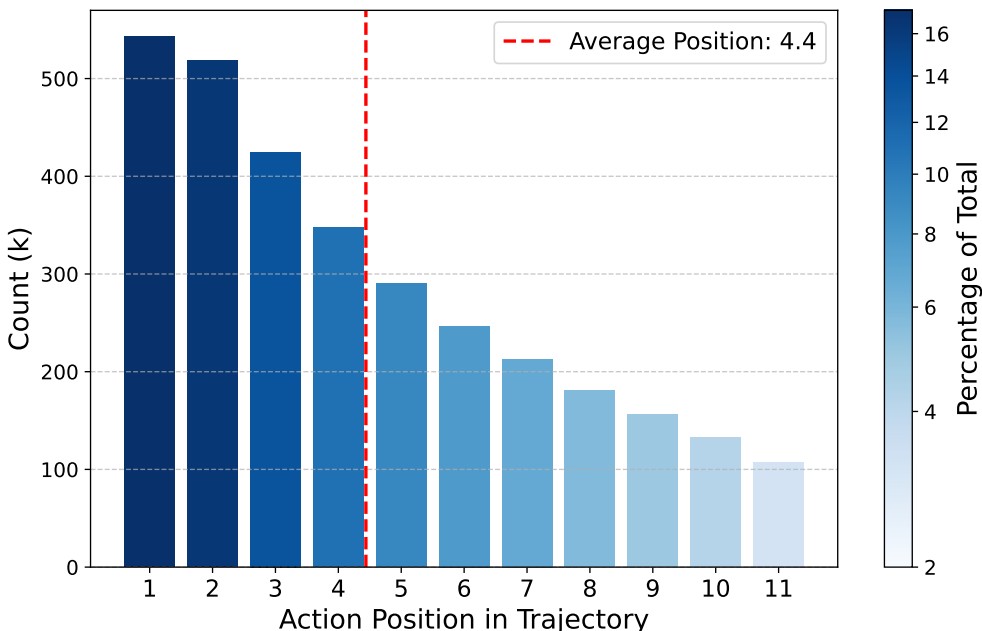

Figure C.1: Distribution of the step at which actions occur within multi-step trajectories. The actions used for training span across different depths, not being limited to initial steps.

$\beta_2 = 0.95$. The learning rate follows a cosine decay schedule with 1000 warmup steps, decaying to 10% of the base LR. We use a global batch size of 192 and enable mixed-precision training.

For the three domain-specific world models (Classifieds, Reddit, and Shopping), we continue training from the Dreamer-7B checkpoint for one epoch using the respective in-domain datasets. These models follow the same training setup, except with a reduced learning rate of `5e-7` and a shorter warmup of 100 steps to ensure stable adaptation on smaller datasets.

# D   Intrinsic Evaluation of World Models

While downstream evaluation with real-world web tasks can provide valuable insights for the model-based planning framework, it involves many other factors (*e.g.*, the policy model) that may influence performance other than world models. Therefore, to rigorously evaluate world models only, we construct an intrinsic evaluation that provides a more controlled and independent assessment while maintaining alignment with the downstream evaluation setting. When training our world models, we use this intrinsic evaluation for model development and checkpoint selection.

## D.1   Dataset Construction

We first sample tasks from Mind2Web (Deng et al., 2023) and manually annotate the trajectories that can complete the tasks as ground truths. Then, we generate deviation trajectories at different states in the ground truth trajectories and assess whether the world model can help distinguish the correct actions from incorrect actions at each state. Specifically, for each state on the ground truth trajectory, we use the ranking model in MindAct (Deng et al., 2023) to select top-5 candidate elements from the webpage that align best with the task intention and current state, excluding the ground truth one. These deviation actions are then extended into full trajectories using a web agent (*i.e.*, SeeAct (Zheng et al., 2024)), automatically evaluated with an LLM-as-a-judge method (Pan et al., 2024a). Since multiple paths may lead to task completion, we filter out deviation actions on successful trajectories to alleviate false negative issue and only retain those in failed

trajectories as negative actions. Using this pipeline, we construct an intrinsic evaluation dataset of 44 tasks, 141 states with deviations, and 279 deviation actions.

## D.2 Evaluation Metrics

For evaluation, we use world models to simulate possible future states for each action and score them using the same scoring function in the planning framework. We then compute three types of metrics: **(1) pair-wise accuracy:** For each pair consisting of a ground-truth action and a deviation action within the same state, if the ground-truth action receives a score greater than or equal to the deviation action, it is counted as correct; otherwise, it is incorrect. Pair-wise accuracy is the proportion of correct pairs across the whole dataset. **(2) state-level accuracy:** Evaluates whether the world model consistently ranks the ground-truth action above all deviations within a given state. A state is considered correct only if the ground-truth action has a score greater than or equal to all its corresponding deviation actions. **(3) task-level accuracy:** Assesses the correctness of an entire task by ensuring that the world model helps correctly select ground-truth action in every state within the task. A task is considered correct only if all its states are correct according to state-level accuracy criteria. We use state-level accuracy as the primary metric to select the best checkpoint, which closely aligns with the downstream task forms.

## D.3 Results

| World Model | Pair-wise Accuracy | State-level Accuracy | Task-level Accuracy |
|---|---|---|---|
| GPT-4o-mini | 85.30 | 78.01 | 47.73 |
| GPT-4o | 87.10 | 80.85 | **52.27** |
| Qwen2-VL-7B | 86.38 | 80.85 | 50.00 |
| Qwen2-VL-72B | 87.10 | 80.14 | 47.73 |
| Dreamer-7B | **88.53** | **82.98** | **52.27** |

Table D.1: Results (%) of various world models on the intrinsic evaluation set.

Table D.1 shows the results of the intrinsic evaluation. Our fine-tuned 7B model achieves comparable performance on task-level accuracy with GPT-4o and even outperforms GPT-4o in terms of pair-wise and state-level accuracy. The scaling trend of training world models can also be observed in intrinsic evaluation as shown in Figure D.1.

For all world models we have evaluated on both intrinsic evaluation and Mind2Web-Live (Pan et al., 2024b), the Pearson correlation coefficient between task-level accuracy and task success rate is 0.8455, indicating a strong correlation between intrinsic evaluation and downstream performance. We hope our intrinsic evaluation can serve as a useful tool for advancing web world model development.

# E More Discussions

## E.1 State Representation and Planning Horizon

In addition to the state change description used in our primary experiments, we explore alternative approaches where GPT-4o predicts either the HTML code or the accessibility tree of the resulting webpage within the simulation. For each of these state representations, we evaluate planning horizons of 1, 2, and 3 steps on VWA benchmark. As depicted in Figure E.1, all three state representations significantly outperform the reactive baseline. However, their effectiveness diminishes as the planning horizon extends to 3 steps, indicating a common limitation in long-horizon simulation across these approaches. Notably, the state change representation exhibits the most pronounced performance degradation as planning horizons extend. This decline is particularly severe with a planning horizon of 3, where performance falls below that of the reactive baseline. This vulnerability stems from its implicit specification of available interactive elements on the current webpage, requiring the model to infer these elements by applying changes to the initial state. In contrast, HTML and accessibility tree representations provide explicit element information. Consequently,

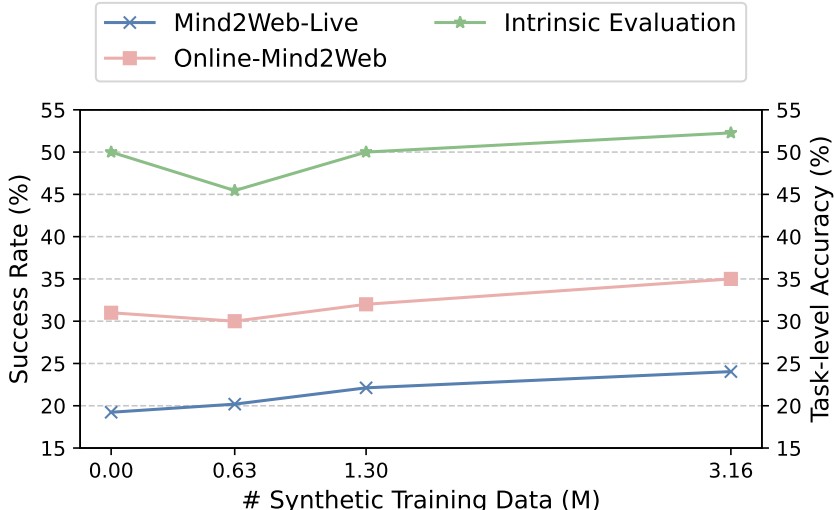

Figure D.1: Performance on two downstream benchmarks and intrinsic evaluation w.r.t. different sizes of training data.

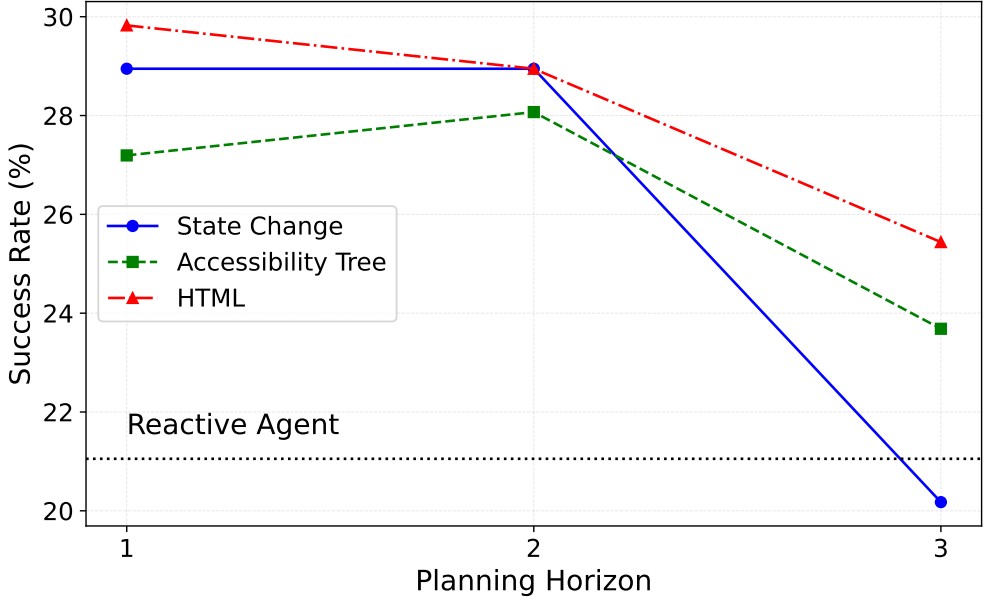

Figure E.1: Performance of WEBDREAMER on the human-verified shopping subset of the VWA dataset, varying both the state representation within simulations and the planning horizon. Planning with long horizon with simulation remains challenging, regardless of the state representation employed.

the state change approach is more susceptible to hallucination during extended simulations. Despite this limitation, the state change approach remains a viable choice given the current capabilities of LLMs. It matches the performance of HTML and accessibility tree representations for planning horizons less than 3 while consuming fewer output tokens.

# F   Case Studies

## F.1   Error Caused by Imperfect World Model Simulation

An error case caused by imperfect world model simulation is shown in Figure F.1.

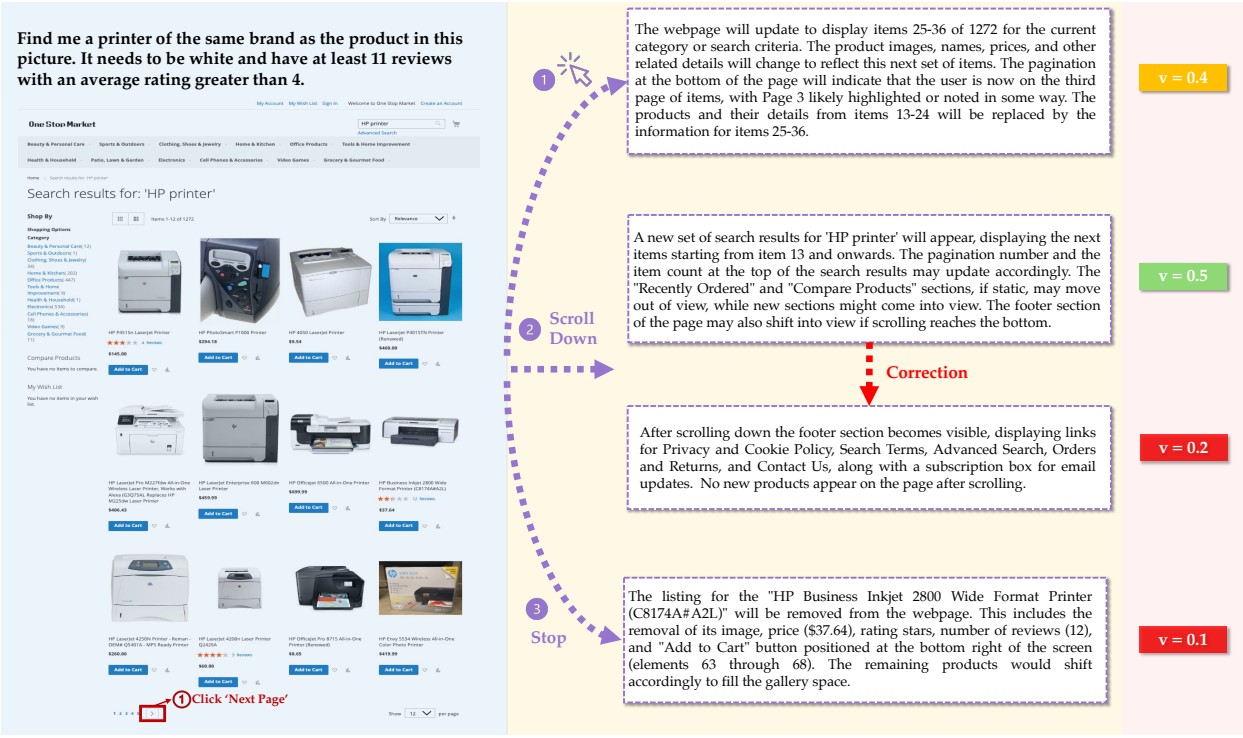

Figure F.1: An error case caused by imperfect world model simulation.

## F.2   Positive Case Benefiting from World Model Simulation

A positive case where the simulation leads to correct action prediction is shown in Figure F.2.

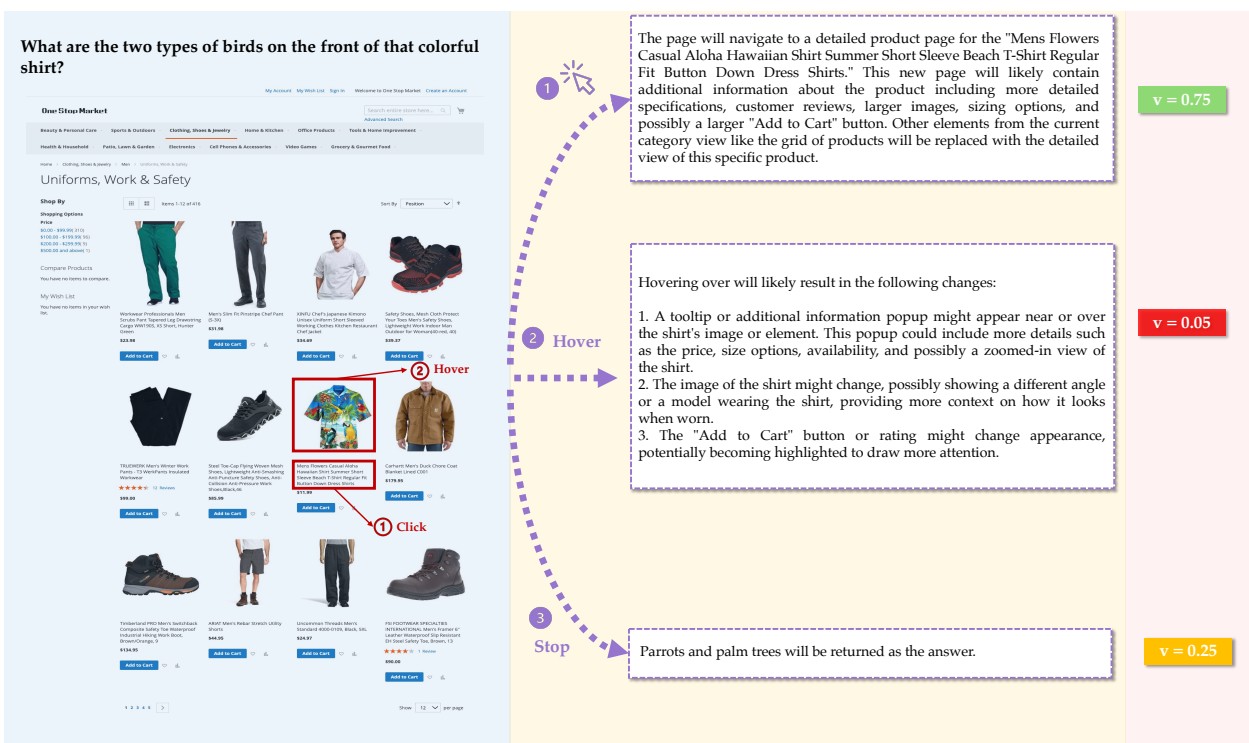

Figure F.2: A positive case where the simulation leads to correct action prediction.

