# OpenReview forum: "Is Your LLM Secretly a World Model of the Internet? Model-Based Planning for Web Agents"
_TMLR — Accepted by TMLR_

### Review · Reviewer_QEPd · 2025-08-03

**Summary Of Contributions:**

This paper introduces a framework, WebDreamer, which applies an LLM to simulate next states and to score simulated trajectories before taking actions. In this framework, the LLM replaces the role of the traditional model (transition kernel) and the value function used in RL, serving as the concept of the World Model.  The proposed method empirically improves the performance over some benchmarks.

Strength: Clear motivation. New idea. Include both training-free and trainable models to validate the performance.

Weakness: Lack of other baselines. The apprach seems to be more general than Web Agents.

**Audience:**

Yes

**Audience Explanation:**

As LLM Agent is a hot topic, I am sure many individuals in TMLR's audience will be interested in knowing the findings of this paper.

**Broader Impact Concerns:**

No broader impact concerns for this paper.

**Claims And Evidence:**

Yes

**Claims Explanation:**

**strengths:**
1. Motivation: Figure 1 clearly explains the motivation of proposing the core idea of this paper, which addresses the existing issues appearing in the Reactive/Tree-Search agents.

2. New idea: While the LLM agent is a crowded field, I definitely believe the same idea has been proposed somewhere by someone. But the specific focus on the WebAgent should be new, and the web agent seems to be more challenging than the purely text-based benchmark.

3. The LLM can be used as the "World Model" in the Web Agent Benchmarks. The concept of world model is apparently abused in the literature; however, in this paper, the world model is clearly described as how the model "Implementation for sim" and "Implementation for score" work, which presents a strong formulation and the fundation to support this claim.

     Continuing the problem setup, this paper investigates how this perspective (LLM as the World Model) helps improve the performance of Web Agent in the standard benchmarks. The experimental results include both the training-free (using GPT-4o) and training approach (using Dreamer-7B), which strongly support the claim made in this paper.


**weaknesses:**

1. Only two baselines are included in the comparison: Reactive and Tree Search. For example, in the VisualWebArena or Online-Mind2Web, it has already included multiple other baselines in their Leaderboard. I can see some baselines are super strong.

    It is not necessary for the proposed method to outperform approaches that are explicitly tailored to this specific task. However, adding some discussion to help readers better understand what other people is doing given the main issue in Reactive or Tree Search.

2. The described approach, "Implementation for sim." + "Implementation for score.", appears applicable to arbitrary tasks. While the authors choose to focus on the Web Agent setting, I am curious: what distinguishes the Web Agent from other tasks? Do the authors believe the same idea (i.e., treating the LLM as the world model) would succeed or fail in other task domains?

**Requested Changes:**

Requested Changes:

* Add more discussions for other approaches in the Web Agent: Explain on how they are related to the web agent strategies (Reactive and Tree Search described in Figure 1). Explain why the World Model approach is potentially better than these approaches.

* Help me understand when the World Model approach will fail. For example, give me an example showing that the LLM cannot  serve as a world model and or a value function in some cases.

---

> ### Author Response · Authors · 2025-08-25
> **Thanks for your encouraging review and thoughtful comments (1/2)**
>
> Thank you for reviewing our work and sharing your thoughtful comments. We are glad you found the motivation and novelty clear, and that our claims are well supported.
>
> ## 1. Limited baseline comparison
>
> > Only two baselines are included in the comparison: Reactive and Tree Search. For example, in the VisualWebArena or Online-Mind2Web, it has already included multiple other baselines in their Leaderboard. I can see some baselines are super strong.
> It is not necessary for the proposed method to outperform approaches that are explicitly tailored to this specific task. However, adding some discussion to help readers better understand what other people is doing given the main issue in Reactive or Tree Search.
> > Add more discussions for other approaches in the Web Agent: Explain on how they are related to the web agent strategies (Reactive and Tree Search described in Figure 1).
>
> Modeling complex web dynamics---diverse content, semantic-rich action space, discrete actions---is highly challenging and remains unexplored. The focus of this work is to answer whether it is possible to repurpose an LLM as a world model of the Internet to facilitate complex web-based tasks. Our baselines were rigorously selected to serve our research question:
>
> (1) Augmenting the standard reactive web agent allows a fair assessment of LLM as a world model of the web.
>
> (2) Our comparison with tree search (when possible) provides deeper insight into LLM’s effectiveness as a world model.
>
> The comparison with these two baselines is already sufficient to demonstrate LLMs’ effectiveness in simulating the web dynamics to facilitate planning (through the comparison with the reactive baseline) with better efficiency (through the comparison with tree search).
>
> While there are other approaches on the leaderboards, they are often focused on other research questions that are orthogonal to our research question (e.g., training a better base model as the policy model). Such methods may not allow a fair, controlled comparison for isolating the effect of using an LLM as a world model.
>
> ## 2. Generalizability of the idea of WebDreamer
> > The described approach, "Implementation for sim." + "Implementation for score.", appears applicable to arbitrary tasks. While the authors choose to focus on the Web Agent setting, I am curious: what distinguishes the Web Agent from other tasks? Do the authors believe the same idea (i.e., treating the LLM as the world model) would succeed or fail in other task domains?
>
> Thanks for raising this question. We agree that the idea of using an LLM as a world model is not confined to web agents and could be applied to other domains, as supported by recent works in planning with LLM-based world models [1]. However, in contrast to simplified and simulated environments (e.g., Blocksworld), the web agent setting presents unique and underexplored challenges:
>
> **Open-ended and dynamic environment**: The Internet evolves continuously, with diverse layouts and ever-changing page content.
>
> **Massive action space**: A single page may contain hundreds of interactive elements, leading to high branching factors in planning.
>
> **Real-world interaction constraints**: Unlike simulated or sandbox environments, search-based planning is often prohibitively slow or infeasible for real-time web interaction.
>
> These factors make model-based planning particularly valuable and challenging in the web agent domain. WebDreamer explores leveraging LLMs as a world model to facilitate planning in such a complex and dynamic environment, and further develops a scalable data synthesis pipeline to train a general world model for the open web.
>
> [1] Hao, Shibo, et al. Reasoning with language model is planning with world model. 2023.

---

> > ### Author Response · Authors · 2025-08-25
> > **Thanks for your encouraging review and thoughtful comments (2/2)**
> >
> > ## 3. Failure Cases
> > > Help me understand when the World Model approach will fail. For example, give me an example showing that the LLM cannot serve as a world model and or a value function in some cases.
> >
> > We provide a concrete error case caused by imperfect world model simulation in Appendix D.1, where the world model incorrectly predicts the consequence of scrolling down, causing it to receive a higher score than the optimal one.
> >
> > Besides, we would like to provide deeper insights into long-horizon simulation failures due to hallucination. The failure mode is consistent: in multi-step simulation, the policy model must propose new actions at each step based on the task and previous simulation results. However, our simulation uses natural language descriptions of state changes, which makes it challenging for the policy model to propose admissible actions at the current step without an actual screenshot. Consequently, the agent tends to hallucinate task-relevant actions even when previous trajectories are incorrect and the proposed action would be unavailable on actual websites. Therefore, as we discussed in the planning horizon section in Sec 5.1, as the planning horizon increases, the trajectories simulated from different actions become less distinguishable to the value function, as they all appear somewhat correct.

---

### Review · Reviewer_mAEQ · 2025-08-05

**Summary Of Contributions:**

This paper presents WebDreamer, a model-based planning framework for web agents that leverages a large language model (LLM) as a world model to simulate the outcomes of web actions prior to execution. The framework demonstrates that both GPT-4o and a domain-tuned model, Dreamer-7B, are capable of simulating web page transitions—such as those resulting from clicking or typing. Empirical results show that Dreamer-7B outperforms both tree search and reactive baselines.

**Audience:**

Yes

**Audience Explanation:**

The main theme of this paper centers on large language models (LLMs), AI agents, and world models—all of which are highly active and timely research topics in the current AI landscape.

**Broader Impact Concerns:**

The ability to simulate website behavior with high fidelity introduces potential risks of misuse, including: modeling phishing infrastructure, automating web scraping or bypassing authentication mechanisms, and exploiting vulnerabilities in interactive systems.
Recommendation: The paper would benefit from a clearer and stronger disclaimer outlining appropriate use boundaries, along with a firm commitment to ethical deployment.

Moreover, the proposed model could ultimately enable autonomous browsing agents capable of acting without user oversight and modifying websites in real time—for instance, submitting forms or making purchases.
Recommendation: The authors should explicitly address safeguards such as requiring user permission, implementing sandboxed environments, or adopting human-in-the-loop mechanisms to ensure transparency, accountability, and user control.

**Claims And Evidence:**

Yes

**Claims Explanation:**

This paper clearly articulates its design and findings. Key design choices—such as planning horizon, scoring functions, and self-refinement—are logically motivated and well justified. The ablation studies effectively demonstrate the impact of planning depth, LLM throughput, and self-refinement. Furthermore, potential limitations, such as the compounding of hallucinations with longer planning horizons, are acknowledged and discussed transparently.

**Requested Changes:**

n/a

---

> ### Author Response · Authors · 2025-08-25
> **Thanks for your positive review and valuable suggestions**
>
> We appreciate the positive assessment of our design choices, ablation studies, and transparent discussion of limitations. We have uploaded the revised submission, with edits marked in blue based on your suggestions.
>
> > The paper would benefit from a clearer and stronger disclaimer outlining appropriate use boundaries, along with a firm commitment to ethical deployment.
>
> Thank you for the thoughtful suggestion. We agree that web agents, especially when powered by world models, may raise potential concerns, and that our released data, while intended as a valuable resource for future research, could be misused if not handled responsibly. In the revised manuscript, we have added an Ethical Considerations section after the conclusion, where we explicitly outline potential risks, safeguards applied during data collection, and the intended research-only scope of our contributions.
>
>
> > The authors should explicitly address safeguards such as requiring user permission, implementing sandboxed environments, or adopting human-in-the-loop mechanisms to ensure transparency, accountability, and user control.
>
> We agree that safeguards are an important direction, and there is emerging research in this space (e.g., [1]). In the revised manuscript, we have added an Ethical Considerations section where we outline potential risks and the intended research-only scope of our work. To further help prevent misuse, we have also included explicit disclaimers and a research-only license with our released data and code, making clear that our contributions are meant solely for academic research under ethical guidelines.
>
> Meanwhile, we note that designing and implementing concrete safeguards such as user permissions, sandboxed environments, or human-in-the-loop mechanisms is largely orthogonal to our technical focus in this paper. Our contribution is to study whether model-based planning with an LLM-based world model can improve the effectiveness and efficiency of web agents. We therefore view the development of practical safeguard mechanisms as an important avenue for future work.
>
> [1] Zheng, Boyuan, et al. WebGuard: Building a Generalizable Guardrail for Web Agents. 2025.

---

### Review · Reviewer_YXpy · 2025-08-11

**Summary Of Contributions:**

This paper presents a framework for model-based planning in web
interaction domains leveraging LLMs for world modeling and control; as
well as a new dataset for fine-tuning the LLM world model. The method
uses bootstrapped value predictions and action pruning / refinement to
avoid expensive inference-time search. Through a series of experiments
in challenging web interaction benchmarks, the proposed method
achieves substantial performance improvements over reactive agents,
and particularly, their 7B-parameter fine-tuned model on their
proposed dataset performs roughly on par with their
framework instantiated with GPT-4o.

**Additional Comments:**

In order to provide some statistical confidence bounds for your results, given how many individual tasks you benchmark on, is it not plausible to use the stratified sampling approach of [1]?

**References**
1. *Deep Reinforcement Learning at the Edge of the Statistical Precipice*. Rishabh Agarwal, Max Schwarzer, Pablo Samuel Castro, Aaron Courville, Marc G. Bellemare. NeurIPS, 2021.

**Audience:**

Yes

**Audience Explanation:**

The paper studies an important problem domain that is at the frontier of agentic LLM research; surely a large portion of the TMLR audience will be interested in the findings of this paper. Moreover, while the paper focuses explicitly on planning for web interaction, the framework it presents can similarly be applied (and be interesting) for general complex planning problems that involve a large multi-modal action space.

**Broader Impact Concerns:**

No broader impact concerns.

**Claims And Evidence:**

Yes

**Claims Explanation:**

Despite certain instances where I felt that terms were not explicitly defined in the main text, largely I am convinced that the claims are accurate and supported by clear evidence. Particularly, the authors included very informative ablation studies to justify the effect of each of the components of their proposed methods. Moreover, they included several control studies / baselines to further paint the picture of which components are crucial for success. I have no major questions or concerns about the results or their correctness.

I also really appreciated the discussion about how the fine-tuned model was evaluated/monitored during training (e.g., via evaluations on a fixed held-out set of tasks).

My one complaint regarding the evidence is that there are no confidence bounds on their empirical results, which makes it difficult to judge statistical significance. However, given the expensive nature of the experiments, perhaps only so much can be done towards this end.

**Requested Changes:**

1.  Nit: the title doesn't feel right to me, particularly w.r.t. "Is
    your LLM Secretly a World Model of the Internet?". Firstly, a
    priori, it's not clear what this question is really asking. Almost
    trivially, an LLM is designed to be a world model of the internet,
    so it seems that the answer should be simultaneously "yes" and "it
    is not even a secret". Do you mean something more precise than this?
    And then, perhaps more importantly, it is also not clear to me
    whether or not your paper answers this question (indeed, I still
    don't know if you intend for the answer to be "yes" or "no").
2.  Nit: In section 3.1, it says "using a learned simulation function
    $\mathtt{sim}(o, a)$". However, $\mathtt{sim}(o, a)$ is not the
    function $\mathtt{sim}$ is. It would be helpful to write instead
    "using a learned simulation function
    $\mathtt{sim}:\mathcal{O}\times\mathcal{A}\to(\mathcal{O}\times\mathcal{A})^*$",
    which (1) is precise, and (2) also tells us the output type of this
    function. Indeed, I originally incorrectly assumed that
    $\mathtt{sim}$ outputs observations, and not trajectories.
3.  Nit: in general POMDPs, I believe $\Omega$ is generally
    non-deterministic, but with your notation $o = \Omega(s)$, it
    appears is deterministic in your setting. This is fine, but perhaps
    it should be stated explicitly.
4.  Similar comment as above for "a scoring function
    $\mathtt{score}(\tau)$" (e.g., the function should just be
    $\mathtt{score}$). This instance is more urgent since the variable
    $\tau$ here has not yet been introduced (so its type is unknown).
5.  In Algorithm 1 (and in the definitions preceding it, shouldn't
    $\mathtt{sim}$ and $\mathtt{score}$ also be conditioned on
    $I$? Particularly in the case of $\mathtt{score}$, I would
    expect the scoring function to highly depend on the instruction
    (whereas perhaps with horizon 1 or 2 as you use, maybe
    $\mathtt{sim}$ is less sensitive…).
6.  Algorithm 1 includes a call to `self_refine`, which isn't explicitly
    discussed in the text. To the left of the pseudocode, there is a
    very brief discussion on self-refinement, but no implementation /
    explicit details are given, unlike in the cases of `sim` and
    `score`. The only such information I could find is with the example
    of the prompt used for self-refinement in Appendix A.2. This should
    be discussed more explicitly in the main text, since you indeed
    found this intervention to be very useful later on in your
    ablations.
7.  Why is the method called WebDreamer? Besides its model-based nature,
    it seems that WebDreamer has effectively nothing in common with the
    Dreamer methods of Hafner et al. (e.g., it is planning in
    observation space, etc).
8.  In section 3.3, you discuss your "scalable data synthesis pipeline".
    This (or at least the data it generated) appears to be a primary
    contribution, but it is not described in much detail. What makes it
    scalable? What does it mean to "\[favor\] frequent interactions like
    clicking while ensuring sufficient coverage of others"? What does it
    mean to "encourage causal dependenies"? I understand what these mean
    intuitively, but how do you *encourage* these things in practice?
    How do you know which web actions are possible at any given point in
    time?
9.  At the end of section 3.3, explain what you mean by "checkpoint
    selection".
10. Excuse me if I missed it, but I see no description of the "reactive"
    baseline. Is this just prompting an LLM (and which one?) for the
    next action given an instruction and current observation, and
    playing it without planning? I see the description of general
    "reactive agents" above, but nothing that says explicitly which
    reactive agent is used in your experiments.
11. In the "effectiveness" subsection of Section 4.2, I don't understand
    how you're coming up with the stated performance gains. For example,
    it says "on the VWA dataset, our proposed method achieves a 34.1%
    relative performance gain". How is that measured?

---

> ### Author Response · Authors · 2025-08-25
> **Thanks for the detailed and constructive feedback (1/3)**
>
> We thank the reviewer for the detailed and constructive feedback. We have uploaded the revised submission, with edits marked in blue based on your suggestions.
>
> ## 1. Title Clarification
>
> > Nit: the title doesn't feel right to me, particularly w.r.t. "Is your LLM Secretly a World Model of the Internet?". Firstly, a priori, it's not clear what this question is really asking. Almost trivially, an LLM is designed to be a world model of the internet, so it seems that the answer should be simultaneously "yes" and "it is not even a secret". Do you mean something more precise than this? And then, perhaps more importantly, it is also not clear to me whether or not your paper answers this question (indeed, I still don't know if you intend for the answer to be "yes" or "no").
>
> Thank you for this question. This title is inspired by the work “Direct Preference Optimization: Your Language Model is Secretly a Reward Model”. What we intended to convey is whether existing LLMs can serve as world models for model-based planning in web environments (i.e., whether they can accurately simulate the causal effects of actions on webpages to enable informed decision-making). The "secretly" refers to whether this capability already exists in the existing LLMs, which is something not explored by the prior literature.
>
> Our paper demonstrates that the answer is largely "yes". LLMs like GPT-4o can serve as effective world models for web planning. However, we also show the limitations, particularly in multi-step simulation.
> Thanks again for the question; we have clarified it in the conclusion in our revision.
>
> ## 2. Mathematical Notation and Formalization Issues
>
> > Nit: In section 3.1, it says "using a learned simulation function \mathtt{sim}(o, a)". However, \mathtt{sim}(o, a) is not the function \mathtt{sim} is. It would be helpful to write instead "using a learned simulation function \mathtt{sim}:\mathcal{O}\times\mathcal{A}\to(\mathcal{O}\times\mathcal{A})^*", which (1) is precise, and (2) also tells us the output type of this function. Indeed, I originally incorrectly assumed that \mathtt{sim} outputs observations, and not trajectories.
> > Similar comment as above for "a scoring function \mathtt{score}(\tau)" (e.g., the function should just be \mathtt{score}). This instance is more urgent since the variable \tau here has not yet been introduced (so its type is unknown).
>
> Thanks for the suggestion on notation precision. We have revised the text to explicitly define the simulation function as $\mathtt{sim}:\mathcal{O}\times\mathcal{A}\to(\mathcal{O}\times\mathcal{A})^*$, which makes clear that it generates imagined trajectories rather than single observations. Similarly, we now denote the scoring function simply as $\mathtt{score}$ (without an undefined $\tau$), clarifying its role in evaluating candidate trajectories.
>
> > Nit: in general POMDPs, I believe \Omega is generally non-deterministic, but with your notation o = \Omega(s), it appears is deterministic in your setting. This is fine, but perhaps it should be stated explicitly.
>
> Thanks for the great catch. We have changed our notation in the updated submission to $o \sim \Omega(s, a)$, making clear that the observation is sampled from the observation function. In fact, in Online-Mind2Web and Mind2Web-Live benchmarks $\Omega$ is non-deterministic, while in the VWA benchmark $\Omega$ is deterministic.
>
> > In Algorithm 1 (and in the definitions preceding it, shouldn't \mathtt{sim} and \mathtt{score} also be conditioned on I? Particularly in the case of \mathtt{score}, I would expect the scoring function to highly depend on the instruction (whereas perhaps with horizon 1 or 2 as you use, maybe \mathtt{sim} is less sensitive…).
>
> Thank you for pointing it out. You are correct - both `sim` and `score` should be conditioned on the instruction I in our formal notation. The scoring function indeed highly depends on the instruction. For the simulation function, while the first module (i.e., state transition prediction, predicting how the webpage changes after an action) is instruction-agnostic, the other one (generating a plausible next action from the predicted state) is instruction-aware.
>
> We have corrected Algorithm 1 to properly reflect these dependencies and edit simulation functions to make it clearer.

---

> ### Author Response · Authors · 2025-08-25
> **Thanks for the detailed and constructive feedback (2/3)**
>
> ## 3. Details of Self-Refinement Step
> > Algorithm 1 includes a call to self_refine, which isn't explicitly discussed in the text. To the left of the pseudocode, there is a very brief discussion on self-refinement, but no implementation / explicit details are given, unlike in the cases of sim and score. The only such information I could find is with the example of the prompt used for self-refinement in Appendix A.2. This should be discussed more explicitly in the main text, since you indeed found this intervention to be very useful later on in your ablations.
>
> Thanks for bringing this up. While we empirically find that the self-refinement step (as a part of the candidate action generation process) improves performance, it represents a straightforward implementation rather than a core methodological contribution. The self-refinement process simply asks the policy model to reflect on the initially generated action candidates and filter out irrelevant or duplicated ones (especially search-related). Given space constraints and our focus on the novel model-based planning process, we chose to prioritize detailed exposition of these core contributions in the main text.
>
> Despite that, we agree with your points on the clarity of this part. We have revised it to include a brief but more explicit description of the self-refinement step in the main text.
>
>
> ## 4. Method Naming (WebDreamer)
> > Why is the method called WebDreamer? Besides its model-based nature, it seems that WebDreamer has effectively nothing in common with the Dreamer methods of Hafner et al. (e.g., it is planning in observation space, etc).
>
> Our choice of "WebDreamer" reflects the core concept of imagination before acting: scoring multiple possible action trajectories through simulation before selecting the optimal one, analogous to how humans "dream up" different scenarios before making decisions.
> The naming honors the broader concept of model-based planning in AI systems, though granted the technical implementation differs significantly from the RL Dreamer methods.
>
>
>
> ## 5. Details of Data Synthesis Pipeline
> > In section 3.3, you discuss your "scalable data synthesis pipeline". This (or at least the data it generated) appears to be a primary contribution, but it is not described in much detail. What makes it scalable? What does it mean to "[favor] frequent interactions like clicking while ensuring sufficient coverage of others"? What does it mean to "encourage causal dependenies"? I understand what these mean intuitively, but how do you encourage these things in practice? How do you know which web actions are possible at any given point in time?
>
> Thank you for pointing this out. We agree that the description of our data synthesis pipeline was too brief in the original submission, and now we have expanded Section 3.3 accordingly in the revised paper. In particular:
>
> **Scalability.**
> The pipeline requires only a pool of seed URLs and runs fully autonomously, without human supervision. It can be parallelized across many workers, enabling the collection of millions of interactions efficiently.
> Favoring frequent interactions.
> We use fixed sampling probabilities that bias toward common operations (e.g., clicks) while still allocating probability mass to less frequent actions such as typing or selecting, ensuring both realism and diversity.
>
> **Encouraging causal dependencies.**
> After each action, we explicitly track HTML changes and prioritize interactions with newly revealed elements (70%; e.g., clicking an item in a dropdown that only appears after hovering). We still allow occasional actions on previously visible elements to maintain variety.
>
> **Action feasibility.**
> At every step, we parse the raw HTML to enumerate valid actions (clickable nodes, text inputs, dropdowns, etc.). We also maintain a mapping between HTML elements and their corresponding positions in the screenshot so that actions are faithfully grounded in the visual state.
>
> These details are now included in the main text, to make the pipeline design clearer. Thanks again for the suggestion.

---

> > ### Author Response · Authors · 2025-08-25
> > **Thanks for the detailed and constructive feedback (3/3)**
> >
> > ## 6. Clarification on Experiment Settings
> > > At the end of section 3.3, explain what you mean by "checkpoint selection".
> >
> > When training the world model, we would like to select the checkpoint yielding the best performance without relying on costly downstream task evaluations for every checkpoint. To address this, we evaluate every 2k steps using our intrinsic evaluation set and select the checkpoint with the highest state-level accuracy (a metric that measures whether the world model can consistently rank the ground-truth action above all deviation actions within a given state). Please see Appendix C for detailed information about the intrinsic evaluation setting.
> >
> > > Excuse me if I missed it, but I see no description of the "reactive" baseline. Is this just prompting an LLM (and which one?) for the next action given an instruction and current observation, and playing it without planning? I see the description of general "reactive agents" above, but nothing that says explicitly which reactive agent is used in your experiments.
> >
> > We appreciate you for raising this point. We have revised Section 4.1 to include more implementation details of baselines.
> >
> > Across all experiments, the reactive agent uses GPT-4o as the policy model. Given the task instruction, current observation, and action history, it predicts the next action without simulation or search. While the LLM may perform internal reasoning (chain-of-thought), we call it reactive because there is no lookahead planning beyond the current state. We use the official GPT+Set-of-Mark reactive implementation from VisualWebArena codebase, and adapt it for Online-Mind2Web and Mind2Web-Live.
> >
> > ##  7. Clarification on Relative Performance Gain
> > > In the "effectiveness" subsection of Section 4.2, I don't understand how you're coming up with the stated performance gains. For example, it says "on the VWA dataset, our proposed method achieves a 34.1% relative performance gain". How is that measured?
> >
> > The relative performance gain is calculated as the percentage improvement over the baseline, which is also used in the tree search baseline paper [1]. Specifically, for the 34.1% relative performance gain, we compare WebDreamer with GPT-4o (23.6% success rate) against the reactive baseline (17.6% success rate): (23.6% - 17.6%) / 17.6% = 34.1% relative improvement.
> >
> > [1] Koh, Jing Yu, et al. Tree search for language model agents. 2024.

---

> > > ### Comment · Reviewer_YXpy · 2025-09-02
> > >
> > > Thanks to the authors for the detailed responses. I still think the title and the name "WebDreamer" could be improved, but these are "small potatoes", as they say. I appreciate the other revisions, these are helpful!

---

> > > > ### Author Response · Authors · 2025-09-09
> > > > **Thanks for the reply**
> > > >
> > > > We are glad that reviewer YXpy found our revisions helpful. Thanks for carefully reviewing our response and updated submission, and for once again providing such detailed and constructive feedback :)

---

### Decision · Action_Editor_enyh · 2025-09-21

**Recommendation:** Accept as is

**Audience:**

Yes

**Audience Explanation:**

The reviewers agree that, though the work is algorithmically fairly straightforward, the empirical findings would be of interest to the TMLR audience.

**Claims And Evidence:**

Yes

**Claims Explanation:**

From the discussion it seems that the paper's claims could be strengthened with more extensive empirical work, the reviewers all agree that the paper provides sufficient evidence for the claims that it does make.